# GNRK: Graph Neural Runge-Kutta method for solving partial differential equations

## Abstract

Neural networks have proven to be efficient surrogate models for tackling partial differential equations (PDEs). However, they may perform poorly or even be inapplicable when deviating from the conditions for which the models are trained, in contrast to classical PDE solvers that rely on numerical differentiation. Striking a balance between efficiency and versatility, this study introduces a novel approach called Graph Neural Runge-Kutta (GNRK), which integrates graph neural network modules with a recurrent structure inspired by the classical solvers. The GNRK operates on graph structures, ensuring its resilience to changes in spatial and temporal resolutions during domain discretization. Moreover, it demonstrates the capability to address general PDEs, irrespective of initial conditions or PDE coefficients. To assess its performance, we benchmark the GNRK against existing neural network based PDE solvers using the 2-dimensional Burgers' equation, revealing the GNRK's superiority in terms of model size and accuracy. Additionally, this graph-based methodology offers a straightforward extension for solving coupled differential equations, typically necessitating more intricate models.

## 1 Introduction

In the fields of science, engineering, and economics, the governing equations are given in the form of partial differential equations (PDEs). Solving these equations numerically for cases where the analytical solution is unkown is a crucial problem. Historically, various types of PDE solvers have been developed. Recently, neural network (NN)-based PDE solvers have been intensively studied in the field of scientific machine learning.

The prototype of an NN solver with a solution ansatz using initial and boundary conditions was first proposed in 1998 (Lagaris et al., 1998). By extending this concept to automatic differentiation in backpropagation, an equation-based model reflecting the PDE, and the initial and boundary conditions of the loss function has been successfully proposed (Raissi et al., 2019). This is a data-free function learning method that maps the PDE domain (position, time, etc.) to a solution function. More recently, an operator learning models that map the initial or source functions of PDEs to the solution functions were proposed (Lu et al., 2019; Li et al., 2019). These are the equation-free, data-based learning methods based on universal approximators of neural networks for operators (Chen & Chen, 1995). In the latter case, myriad of variants have been proposed from the view point of operator parameterization (Shin et al., 2022) and geometry adaptation (Li et al., 2022; 2023). The background of NN solvers is well summarized in the survey paper (Huang et al., 2022).

Owing to these successful studies, recent NN solvers have exhibited remarkable performance and have become time-efficient surrogate models through inference. However, the performance of these models are mostly validated under simple experimental situations, and it is still difficult to robustly train NN solvers when the conditions of PDEs, such as the initial condition, PDE coefficients and spatiotemporal mesh change with the data.

Unlike NN solvers, classical solvers such as finite-element method (FEM), finite-difference method (FDM), and finite-volume method (FVM) can find the solution of the PDE regardless of its form, initial and boundary conditions, given an exact PDE. There is a trade-off in terms of versatility and usability between classical solvers utilizing analytical techniques and NN solvers relying on expressive data-based NN training. We can naturally assume that a hybrid model that appropriately

utilizes the advantages of each scheme will be a characteristic of the ideal PDE solver. Several attempts have been made to apply classical solver schemes to NN solvers. In Karlbauer et al. (2022), the changes in the initial and boundary conditions were addressed by applying a FVM scheme to physics-informed neural networks (PINNs). In Brandstetter et al. (2022), a message passing neural operator model that can train changes in PDE coefficients and boundary conditions in spatial mesh data was proposed.

Meanwhile, coupled differential equations with a graph spatial domain rather than the usual Euclidean space have not received much attention. The coupled systems with numerous interacting elements exhibit rich sets of properties. This includes the effect of underlying topology on the solution in addition to the conventional PDE conditions. No solid model for coupled differential equations and coupled systems has been extensively addressed.

In this study, we developed a Graph Neural Runge-Kutta (GNRK) method that combines the RK method, a representative classical solver, with graph neural networks (GNNs).

1. In the GNRK, the RK method is reconstructed as a recurrent structure with residual connections, and the governing function is approximated by a GNN.

2. The proposed method does not depend on spatial as well as temporal discretization and can robustly predict the solution despite changes in the initial conditions and PDE coefficients.

3. This method can control the prediction accuracy by adjusting the RK order given by the recurrent depth.

To the best of our knowledge, the GNRK is the first model that can solve a PDE regardless of the initial conditions, PDE coefficients, and the spatiotemporal mesh. It can even predict solutions with higher precision than the training data by modifying its' structure without retraining or fine-tuning. We verified that the GNRK predicts PDE solutions superiorly compared to the representative NN solvers of equation-, neural operator-, and graph-based algorithms in challenging scenarios, where the above conditions vary in the two-dimensional (2D) Burgers' equation. In addition, the GNRK is applied to three different physical systems governed by coupled ordinary differential equations (ODEs), each of which has linear, nonlinear, and chaotic properties. In addition to the robustness of the conditions discussed in the Euclidean space, we confirmed that the GNRK accurately predicts solutions irrespective of topological changes in the graph structure.

## 2 GRAPH NEURAL RUNGE-KUTTA

In this section, we describe the GNRK, a new data-driven approach for solving PDEs. This is a hybrid approach that combines the well-studied methods of numerical PDE solvers with the powerful expressiveness of artificial neural networks, particularly the GNN.

First, the following PDEs are considered:

$$\frac{\partial s(\boldsymbol{x}, t)}{\partial t} = f(s; C). \tag{1}$$

A dynamic system with state $s(\boldsymbol{x}, t)$ evolves over time according to the governing equation $f$ with constant coefficients $C$. The PDE is defined in a spatial domain $\Omega$ with the boundary condition (BC) specifying the constraint of $s(\boldsymbol{x}, t)$ at the boundary $\boldsymbol{x} \in \partial\Omega$. The BC can take different forms; for example, when no condition is given, an open BC is assumed, and when boundaries are connected, a periodic BC is used. The time domain of interest is denoted as $\mathcal{T} \equiv [0, T]$.

### 2.1 CLASSICAL SOLVER

**Temporal discretization.** Because it is impossible to compute continuous values, most classical solvers are based on discretization of both $\Omega$ and $\mathcal{T}$. For discretization over $\mathcal{T}$, the time interval $\Delta t$ is chosen for each time step. In the PDE given in equation 1, the state $s(t + \Delta t)$ depends only on $s(t)$, which is the state at the immediate previous time. Although it is common to have a constant $\Delta t$ over time, we use a nonuniform discretization of $\mathcal{T}$ with time steps $\{\Delta t_k\}_{k=1}^{M}$ where $M$ is the number of time steps.

**Euclidean space discretization.** When the spatial domain $\Omega$ is a Euclidean space, it is often discretized into a square grid. Similar to discretization over $\mathcal{T}$, a uniform square grid is often used, in which all grid points are equally spaced. We choose a more general, nonuniform square grid that does not change over time but has a different spacing between grid points, as shown in Fig. 1(a). Assuming that $f$ is differentiable or localized over space, the state at grid point $i$ is updated using only the states of the neighboring points $\mathcal{N}(i)$ in the grid.

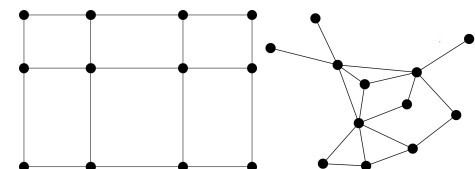

Figure 1: Discretization of spatial domain $\Omega$. (a) 2D Euclidean spatial domain discretized by a nonuniform square grid. The spacing between grid points can be different. (b) Graph spatial domain, represented by nodes and edges. Both (a) and (b) consist of 12 nodes and 17 edges but have different topologies.

**Graph spatial domain.** We also consider $\Omega$ to be the graph spatial domain. As shown in Fig. 1(b), a graph domain consisting of nodes and edges is discretized by itself. Similar to the Euclidean spatial domain, we assume that the state of node $i$ depends directly only on the state of its neighboring nodes $\mathcal{N}(i)$. However, there are some crucial differences between the graph spatial domain and the Euclidean space. First, each node is not fixed in the coordinate system; therefore no metric defines the distance or direction between the nodes. In addition, because there are no boundaries in the graph, the boundary condition is absent. Regarding the connection pattern of the nodes, degree $|\mathcal{N}(i)|$ is defined as the number of neighbors of node $i$. Grids can be considered regular graphs where all nodes have the same degree, with special care required for nodes on $\partial\Omega$ depending on the BC. Hereafter, we consider an arbitrarily discretized $\Omega$ as a graph $\mathcal{G}$.

**Runge-Kutta method.** Among the many classical solvers, we use the explicit Runge-Kutta method, an effective and widely used method proposed by Runge (1895), and later developed by Kutta (1901). This algorithm has an order $m$ that sets the trade-off between the computation and accuracy of the result. For node $i$, the $m$-th order explicit Runge-Kutta method ($RK^m$) updates $s(\boldsymbol{x}_i, t)$ to the next state $s(\boldsymbol{x}_i, t + \Delta t)$ using $m$ different intermediate values $\{\boldsymbol{w}_i^l\}_{l=1}^m$ as

$$s(\boldsymbol{x}_i, t + \Delta t) = s(\boldsymbol{x}_i, t) + \Delta t \sum_{l=1}^m b^l \boldsymbol{w}_i^l. \tag{2}$$

Assuming the spatially localized $f$, each $\boldsymbol{w}_i^l$ is computed in parallel for $i$ and recurrently for $l$ as follows:

$$\begin{aligned}
\boldsymbol{w}_i^1 &= f(\mathbb{S}_i; C) \\
\boldsymbol{w}_i^2 &= f(\mathbb{S}_i + a_{2,1}\mathbb{W}_i^1 \Delta t; C) \\
&\vdots \\
\boldsymbol{w}_i^m &= f(\mathbb{S}_i + \left[a_{m,1}\mathbb{W}_i^1 + \cdots + a_{m,m-1}\mathbb{W}_i^{m-1}\right]\Delta t; C).
\end{aligned} \tag{3}$$

For simplification, we abuse the set notation $\mathbb{S}_i \equiv \{\boldsymbol{s}_j | j \in \{i\} \cup \mathcal{N}(i)\}$ and $\mathbb{W}_i^l \equiv \{\boldsymbol{w}_j^l | j \in \{i\} \cup \mathcal{N}(i)\}$ where $\boldsymbol{s}_i \equiv s(\boldsymbol{x}_i, t)$. The coefficients of the RK method, $a$ and $b$, can be selected from the Butcher tableau (Butcher, 1964).

## 2.2 Architecture of GNRK

**Recurrent structure with residual connection.** There have been many approaches for understanding classical solvers as neural networks with residual connections, especially those that consider the entire $RK^m$ step to be a single layer (Lu et al., 2018; Queiruga et al., 2020). However, this leads to the problem of optimizing the deep network, which was addressed using the adjoint method (Chen et al., 2018). By contrast, the neural network $f_\theta$ in the GNRK approximates the governing equation $f(s; C)$ where $\theta$ denotes trainable parameters. As shown in Fig. 2(a), $RK^m$ is reproduced by the recurrent structure of depth $m$ (Rico-Martinez et al., 1992) using the approximator $f_\theta$ with the residual connections. The coefficients of the element-wise linear combinations at the residual connections are defined by the Butcher tableau and $\Delta t$. Note that $f_\theta$ is shared across the $m$ sub-steps of $RK^m$ as shown in equation 3; therefore, a fixed size of GNRK can be used for different $m$.

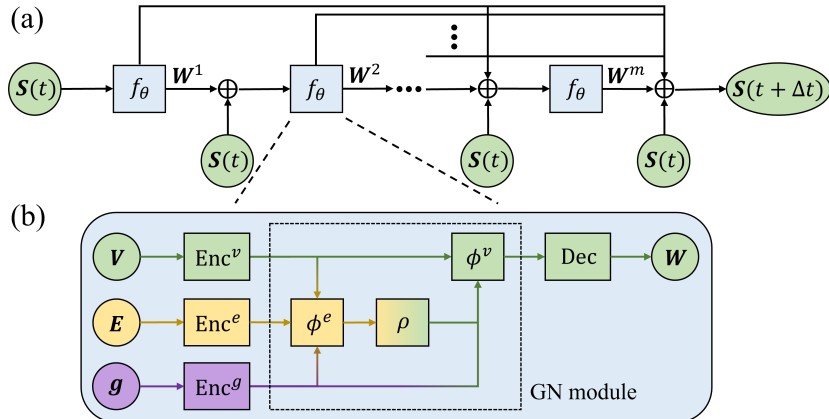

Figure 2: The full structure of the GNRK. (a) The RK$^m$ step is reproduced by the recurrent structure of depth $m$ with residual connection. Unlike the traditional residual connection, the results of all previous blocks are used for the next step. (b) Typical structure of GNN $f_\theta$ with three different types of input features: node $\boldsymbol{V}$, edge $\boldsymbol{E}$, and global features $\boldsymbol{g}$ represented by colors.

**Graph Neural Network.** Consider the first substep of equation 3, which computes $\boldsymbol{W}^1 \equiv [\boldsymbol{w}_1^1, \ldots, \boldsymbol{w}_N^1]$ from $\boldsymbol{S} \equiv [\boldsymbol{s}_1, \ldots, \boldsymbol{s}_N]$. Neural network structures, including multi-layer perceptrons (MLPs), are capable of this operation; however, GNNs are among the most efficient (Iakovlev et al., 2021). Similar to RK$^m$, it computes $\boldsymbol{w}_i^1$ in parallel using only the nodes and their neighbors ($\mathbb{S}_i$) by sharing the parameters for all nodes. In addition to efficiency, another benefit of GNNs is that they can handle generic graphs where each node has a different degree and even domains with a different number of nodes and edges. The typical structure of $f_\theta$ used in the GNRK is shown in Fig. 2(b), following the graph network (GN) framework (Battaglia et al., 2018).

**Inputs.** The inputs to the GNN are categorized into three types: node features $\boldsymbol{V} \equiv [\boldsymbol{v}_i]^T$, edge features $\boldsymbol{E} \equiv [\boldsymbol{e}_{i,j}]^T$, and global features $\boldsymbol{g}$. The state of each node $\boldsymbol{s}_i$ or the linear combination of $\boldsymbol{s}_i$ and $\{\boldsymbol{w}_i^l\}$ are typical examples of node features, because the state of the system is assigned to the node. Note that all input features except the state are fixed, whereas $f_\theta$ is evaluated $m$ times along the recurrent structure.

**Encoder.** Each input feature is embedded into a higher dimension by a 2-layer MLP encoder. Borrowing the parameter sharing property of the GNN, we share the node encoder Enc$^v$ across all the node features $\boldsymbol{v}_i^{\text{emb}} = \text{Enc}^v(\boldsymbol{v}_i)$, $^\forall i$. Similarly, all the edge features $\boldsymbol{e}_{i,j}$ are embedded by sharing Enc$^e$. As only a single $\boldsymbol{g}$ exists per graph, Enc$^g$ is not shared.

**Graph Network Module.** The GN is a framework that explains the operations on a graph and covers popular message-passing neural networks (MPNNs) (Gilmer et al., 2017). It can describe a wide variety of operations on the graph, including updating edge and global features that the MPNN cannot address. As shown in the Fig. 2(b), the GN module used in GNRK is computed as follows:

$$\boldsymbol{e}_{ij}' = \phi^e(\boldsymbol{v}_i, \boldsymbol{v}_j, \boldsymbol{e}_{ij}, \boldsymbol{g}), \quad \bar{\boldsymbol{e}}_i' = \rho(\boldsymbol{e}_{ij}', {}^\forall j \in \mathcal{N}(i)), \quad \boldsymbol{w}_i = \phi^v(\bar{\boldsymbol{e}}_i', \boldsymbol{v}_i, \boldsymbol{g}). \tag{4}$$

Initially, the edge features are updated in parallel by sharing the per-edge update function $\phi^e$. Then, $\boldsymbol{e}_{ij}'$ is aggregated into the corresponding nodes by a permutation-invariant operation $\rho$ such as the sum or average. Finally, the node features are updated in parallel by sharing the per-node update function $\phi^v$. The two update functions $\phi^e, \phi^v$ and one aggregation function $\rho$ can be designed in various ways, and the appropriate form depends on the given PDE $f$.

**Decoder.** After the GN module, we use a 2-layer MLP decoder to output $\boldsymbol{W}$. Like the encoding process, one decoder is shared over all updated node features.

**Optimization.** For a given dynamic system, a trajectory is defined as a collection of states that change over time $[s(t_0), s(t_1), \ldots, s(t_M)]$. The GNRK is trained by a one-step prediction: for given $s(t)$, minimize the Mean Squared Error (MSE) between the next state $s(t + \Delta t)$ and the prediction $\tilde{s}(t + \Delta t)$. Because the recurrent structure is limited to depth $m$, the backprapagation is tractable.

Furthermore, most of the trainable parameters for $f_\theta$ are shared across nodes or edges, allowing data-efficient training. Note that training the GNRK does not require prior knowledge of the governing equation $f$. After optimization, the model predicts entire trajectory in a rollout manner. Starting from the given initial condition $\tilde{s}(0) \equiv s(0)$, it recursively generates $\tilde{s}(t + \Delta t)$ using $\tilde{s}(t)$ as input.

## 3 GENERALIZATION CAPABILITY

For a given dynamic system consisting of a PDE $f$ and domain $\Omega, \mathcal{T}$, many conditions exist which determine the solution. In this section, we classify the criteria into the following four categories, and provide intuitions on why GNRK can be generalized to each of them without retraining.

- Initial conditions

- PDE coefficients

- Spatial discretization

- Temporal discretization and the order of RK

From a practical standpoint, a good PDE solver should deliver a robust performance across a wide range of criteria. In other words, even when unseen values for each criteria are probided, the solver should accurately infer the trajectory. The previously proposed NN solvers are only applicable to a very limited subset of the above regimes or show poor performance even when they are applicable.

**Initial conditions.** The initial condition $s(0)$ is the state in which a dynamic system starts and solving the PDE with different initial conditions is a routine task. The GNRK excels at this task because it uses localized information to predict the next state. Even for different input states, if they have common values in any localized space, the computations in that region are the same. This allows the GNRK to learn various local patterns using few initial conditions, leading to data-efficient training.

**PDE coefficients.** Depending on the constant coefficient $C$ in the governing equation $f$, the system may exhibit different trajectories. We categorize $C$ into three types, node, edge, and global coefficients, in line with the input of the GNN $f_\theta$. Unlike most NN solvers that do not take $C$ as the input, the GNRK is designed to work with different $C$. Similar to the case of state, with parameter sharing, the GNRK learns the effect of $C$ in various transitions from a single data sample. The other NN solvers can be modified to allow the model to use $C$ as input; however, such efficient training is not possible.

**Spatial discretization.** For the Euclidean spatial domain, the discretization of space $\mathcal{G}$ affects the accuracy of the trajectory, and different discretizations are commonly considered depending on the application. Because the GNRK can be applied to a nonuniform square grid, it is possible to estimate the state at an arbitrary position. The position of each node in Euclidean space can be encoded as node or edge features: coordinates assigned as node features or the relative positions between two neighboring nodes as edge features. If the governing equation depends directly on the coordinates, the node encoding strategy is appropriate; otherwise, both are valid. In the graph spatial domain, the role of $\mathcal{G}$ becomes much larger because the trajectories can vary dramatically depending on the connectivity structure of $\mathcal{G}$ and the numbers of nodes and edges.

**Temporal discretization and the order of RK.** Similar to spatial discretization in the Euclidean spatial domain, the temporal discretization $\{\Delta t_k\}_{k=1}^M$ can vary with the quality of the trajectory. By using $\Delta t$ of different sizes, the GNRK can make predictions at arbitrary times. In the GNRK, both $\{\Delta t_k\}_{k=1}^M$ and order $m$ are captured in the recurrent structure, as shown in Fig. 2(a), leaving trainable $f_\theta$ independent. Therefore, the GNRK model can be tuned to have a different $m$ without retraining and can make predictions using unseen $\Delta t$. This property is particularly beneficial when only low-quality data are available for training and a more precise trajectory is required for deployment. The method of improving accuracy is most evident in chaotic systems, where the trajectories are very sensitive to numerical precision.

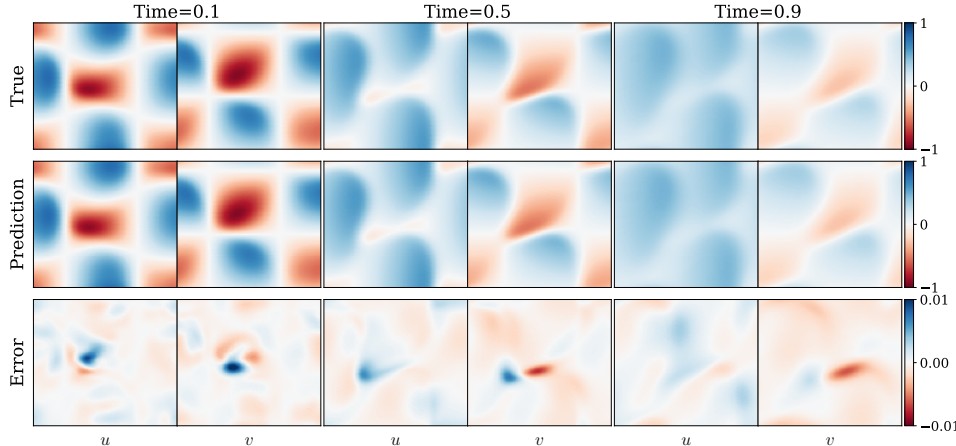

Figure 3: Three snapshots of the trajectory of the 2D Burgers' equation, GNRK prediction, and its error. The error is defined as the difference between the true and predicted trajectory, and is plotted in zoomed-in colormaps for visibility.

## 4 EXPERIMENT

### 4.1 EUCLIDEAN SPATIAL DOMAIN

**2D Burgers' equation.**

$$\frac{\partial u}{\partial t} = -u\frac{\partial u}{\partial x} - v\frac{\partial u}{\partial y} + \nu\left(\frac{\partial^2 u}{\partial x^2} + \frac{\partial^2 u}{\partial y^2}\right), \quad \frac{\partial v}{\partial t} = -u\frac{\partial v}{\partial x} - v\frac{\partial v}{\partial y} + \nu\left(\frac{\partial^2 v}{\partial x^2} + \frac{\partial^2 v}{\partial y^2}\right). \quad (5)$$

We consider a 2D Burgers' equation which is given as a coupled nonlinear PDE. It describes the convection-diffusion system for a fluid of viscosity $\nu$ with velocities of $u(x, y; t), v(x, y; t)$ in each $x$ and $y$ direction, respectively. The spatial domain is selected as $\Omega \equiv [0, 1]^2$ with a periodic BC, and the time domain is $\mathcal{T} \equiv [0, 1]$. The method for numerical differentiation on nonuniform square grid is described in Appendix. A.

To demonstrate the robustness of NN solvers over each criteria, we build the following four datasets. *Dataset I* consists of the trajectories starting from different initial conditions. *Dataset II* varies the PDE coefficient $\nu$ over a limited range, and *Dataset III* tests spatial invariance by using nonuniform square grids of different sizes. *Dataset IV* is composed of samples created with nonuniform $\{\Delta t_k\}_{k=1}^{M}$. In addition, the trajectory quality for training and testing is set differently by changing the Runge-Kutta order $m$ to 1 and 4, respectively. Detailed settings can be found in Appendix. B. Across all the datasets, we generate a limited number of 20 training samples, and 50 test samples to evaluate the model.

Figure 3 shows a successful prediction of the GNRK for a trajectory in *Dataset I*. The predicted trajectory captures the characteristics of the Burgers' system; the initial formation of shockwaves and their subsequent dissipations owing to diffusion.

The mean absolute error (MAE) between the true and predicted trajectories is used to quantify the predictive performance of the models. Figure 4 shows the evolution of MAE for the test data from the four datasets on a logarithmic scale. The errors are averaged over all nodes in the spatial domain and two states, $u$ and $v$. Becase the predictions of GNRK are made in a rollout manner, the errors tend to accumulate over time. However, the errors are consistent across samples, indicating that the GNRK is robust in all cases, and not only for test data that are similar to some of the trained data. We compare the results with those of the GraphPDE proposed in Iakovlev et al. (2021). Similar to the GNRK, GraphPDE is one of the few models that can be trained and tested on all four datasets.

In Table 1, we report the performances of the GNRK along with those of other NN solvers. PINN (Raissi et al., 2019) maps the spatiotemporal domain directly to the solution of the PDE. This makes it impossible to make predictions for different initial conditions and PDE coefficients.

|  | PINN | FNO* | GNO* | GraphPDE | GNRK (ours) |
|---|---|---|---|---|---|
| Dataset I | - | $3.46 \times 10^{-2}$ | $1.65 \times 10^{-1}$ | $1.24 \times 10^{-1}$ | $\mathbf{1.04 \times 10^{-3}}$ |
| Dataset II | - | $1.24 \times 10^{-3}$ | $5.58 \times 10^{-3}$ | $7.53 \times 10^{-2}$ | $\mathbf{1.13 \times 10^{-3}}$ |
| Dataset III | $2.72 \times 10^{-1}$ | - | $5.59 \times 10^{-2}$ | $7.89 \times 10^{-2}$ | $\mathbf{4.56 \times 10^{-3}}$ |
| Dataset IV | $2.58 \times 10^{-1}$ | - | - | $6.05 \times 10^{-2}$ | $\mathbf{1.44 \times 10^{-3}}$ |
| Number of parameters | 17,344 | 2,376,610 | 550,658 | 21,602 | $\mathbf{10,882}$ |
| Wall clock (second) | $0.593 \pm 0.003$ | $1.76 \pm 0.01$ | $5.16 \pm 0.05$ | $\mathbf{0.34 \pm 0.08}$ | $7.32 \pm 0.10$ |
| GPU memory (MiB) | $\mathbf{122}$ | 609 | 317 | 237 | 153 |

Table 1: Mean absolute errors of test datasets using various NN solvers. The results marked with - are cases where training is impossible due to the numerical limitations of corresponding PDE solvers. *For FNO and GNO, we use 100 train samples to obtain comparable results to GNRK. The wall clock is $1.525 \pm 0.006$ second, when using a classical numerical solvers.

In contrast, Fourier neural operator (FNO) (Li et al., 2019) approximates the kernel integral operator which maps the function $s(t)$ to the next time step $s(t + \Delta t)$. Since fast Fourier transform is used, it only applies to uniform spatial grids and fixed temporal intervals. As a member of the same neural operator architecture, graph neural operator (GNO) (Anandkumar et al., 2019) approximates the kernel integral with the GNN, which can cover nonuniform grids. GraphPDE (Iakovlev et al., 2021) uses a MPNN which approximates the governing equation, and NeuralODE to compute the time evolution of the trajectory. This allows the GraphPDE to be applicable to all four datasets, but its performance is worse than the GNRK. Employing the least number of trainable parameters, the GNRK shows the best performance across all datasets. The detailed implementation of each model can be found in Appendix B.

## 4.2 GRAPH SPATIAL DOMAIN

In this section, three physical systems governed by coupled ODEs defined in the graph spatial domain are considered. Rather than examining the generalization domain of the GNRK individually, as in the Euclidean spatial domain, we use a dataset in which all the four criteria are different. In particular, for the spatial domain $\mathcal{G}$, we used three random graph models to investigate the effect of the graph topology. They are distinguished by the homogeneity of the degree distributions, where the most homogeneouse being the random regular (RR) graph, where all nodes have the same degree, similar to a grid. The next is the Erdős-Rényi (ER) graph (Erdős & Rényi, 1960), where the degree distribution follows a Poisson distribution, and the last is the Barabási-Albert (BA) graph (Barabási & Albert, 1999), which is the most heterogeneous as the degree distribution follows a power-law.

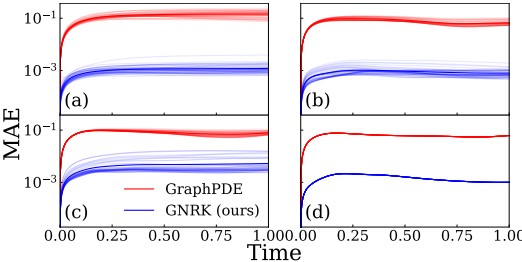

Figure 4: The mean absolute error (MAE) over time for (a) *Dataset I*, (b) *Dataset II*, (c) *Dataset III*, (d) *Dataset IV* in logarithmic scale. The light colored lines are the results for 50 test samples, and the dark colored lines are the average over the light lines.

**Heat equation.**

$$\frac{dT_i}{dt} = \sum_{j \in \mathcal{N}(i)} D_{ij}(T_j - T_i). \tag{6}$$

First, we consider the heat equation that describes the diffusion system. The heat equation defined in Euclidean space is a PDE, $\partial_t T = \nabla^2 T$; however, in the graph spatial domain, it is replaced by a coupled linear ODE as follows. Each node has a temperature $T_i$, and two nodes are connected by an edge with a dissipation rate $D_{ij}$ exchanging heat.

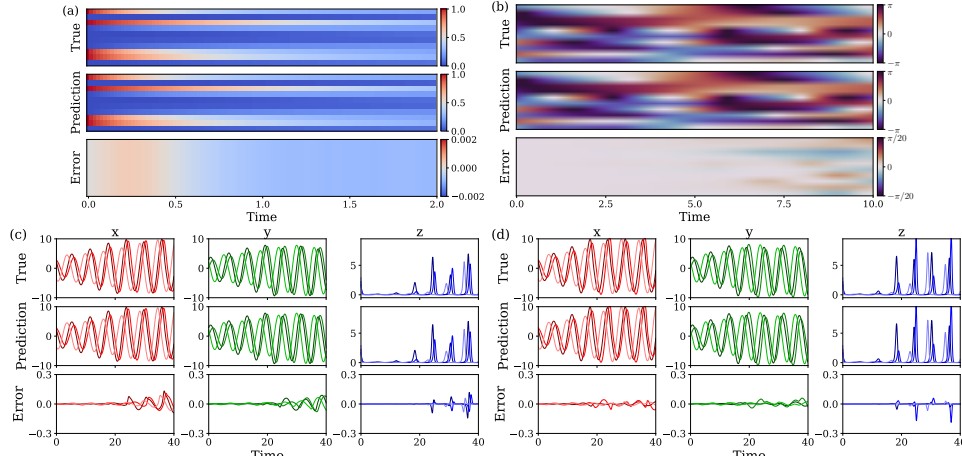

Figure 5: Results from the GNRK model for test samples in (a) heat, (b) Kuramoto, and (c),(d) coupled Rössler systems. Colors indicate the temperature $T_i \in [0, 1]$ in the heat system, and the phase $\theta_i \in (-\pi, \pi]$ in the Kuramoto system. For the coupled Rössler system, 3D coordinate values $(x_i, y_i, z_i)$ are displayed, with nodes distinguished by line brightness. Only the 10 nodes (heat, Kuramoto) or 3 nodes (coupled Rössler) with the largest errors from the test samples are shown. (c) and (d) depict trajectories of the same coupled Rössler system, differing in numerical precision, with $m = 4$ in (c) and $m = 1$ in (d).

**Kuramoto equation.**

$$\frac{d\theta_i}{dt} = \omega_i + \sum_{j \in \mathcal{N}(i)} K_{ij} \sin(\theta_j - \theta_i). \tag{7}$$

The Kuramoto equation is a coupled nonlinear ODE, which governs a dynamic system of coupled oscillators. Their phase $\theta_i \in (-\pi, \pi]$ evolves as a result of its natural angular velocity $\omega_i$ and nonlinear interactions with neighbors $j$ with the magnitude of the coupling constant $K_{ij}$. It is well known that a large $K_{ij}$ can lead the system to a synchronized state in which all the oscillators have the same phase. Therefore, we take a relatively small value of $K_{ij}$ to ensure that the system has diverse trajectories without synchronization.

**Coupled Rössler equation.**

$$\frac{dx_i}{dt} = -y_i - z_i, \quad \frac{dy_i}{dt} = x_i + ay_i + \sum_{j \in \mathcal{N}(i)} K_{ij}(y_j - y_i), \quad \frac{dz_i}{dt} = b + z_i(x_i - c). \tag{8}$$

Finally, we consider the dynamical system of Rössler attractors with pairwise attractive interactions. The nodes are attractors in a three-dimensional space with coordinates $(x_i, y_i, z_i)$ coupled on the $y$-axis. Like the Kuramoto system, we choose small values of $K_{ij}$ to prevent synchronization, with all nodes' positions evolving in a similar fashion. Furthermore, we deliberately select the values of $a, b$ and $c$ such that the system exhibits chaotic behavior. The choice of these coefficients makes the prediction of a coupled Rössler system one of the most challenging tasks encountered by a PDE solver.

The detailed configurations of the datasets for each system and the implementation of the GNRK can be found in Appendix. C. We plot the results of the GNRK for test samples for the three systems in Fig. 5. Because each system consists of dozens to hundreds of nodes, it is impossible to draw the trajectories of all nodes; therefore, we select few nodes with the largest error. Across all systems, despite this complex inference task, the errors from the true trajectories are so small that they could only be visible on a zoomed-in scale.

**Precision boosting.** The capability of producing different precisions by tuning $m$ is particularly noticeable in the coupled Rössler system, which has chaotic properties. The first rows of Fig 5(c) and (d) show the two trajectories from the same condition but with different numerical precisions: $m = 4$ and $m = 1$, respectively. Over time, errors from small discrepancies in numerical precision lead to

different trajectories, which are evident in $z$-axis. Nevertheless, with the proper tuning of $m$, the GNRK can predict both trajectories without significant errors. Similar precision-dependent results are observed for the other two systems. The top row in Fig. 6 shows the MAE when the prediction is made while maintaining training precision ($m = 1$). This shows 10x larger error compared with the result after tuning the GNRK to $m = 4$ shown in the second row in Fig. 6.

**Impact of graph topology.** Recall that the connectivity pattern or topology of the graph significantly affects the trajectory. Therefore, we investigate the effect of the topology of the underlying graph on the performance of the GNRK. The bottom three rows in Fig. 6 show the MAE obtained from samples with underlying graphs of random regular (RR), Erdős-Rényi (ER), Barabási-Albert (BA) graph, respectively. The second row shows the MAE of the total number of samples, which is the average of the bottom three rows. We did not observe any significant performance differences based on the topology.

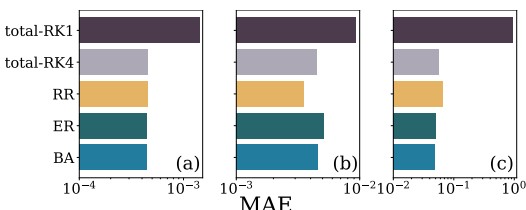

Figure 6: Mean Absolute Error (MAE) of (a) heat, (b) Kuramoto, and (c) coupled Rössler systems on a logarithmic scale. The error statistics are computed for different underlying topology of random regular (RR), Erdős-Rényi (ER), Barabási-Albert (BA) graph, respectively, and the average of them is shown in the total row. RK1 and RK4 represent the first and the 4-th order Runge-Kutta (RK) methods, respectively.

## 5 CONCLUSION

In this study, we propose the Graph Neural Runge-Kutta (GNRK), a novel approach for solving differential equations that combines the advantages of versatile classical solvers and expressive artificial neural networks. The GNRK is a highly capable PDE solver because it can predict solutions under different initial conditions and PDE coefficients. Furthermore, it is invariant to spatial and temporal discretization, allowing one to obtain a solution at arbitrary spatial coordinates and times. Finally, the precision of the solution generated by the GNRK can exceed that of the trained data without any additional training.

These advantages over other NN-based PDE solvers arise from the structure of GNRK. By reproducing the $m$-th order RK as a recurrent structure of depth $m$ with residual connections, the GNRK can accommodate an arbitrary order $m$. The recurrent structure also makes the GNRK invariant to arbitrary temporal discretization using $\Delta t$ as the coefficient of residual connections rather than the input of the neural network. We choose a GNN as the recurrent module $f_\theta$, which approximates the governing differential equation $f$. The implementation of GNN provides the robustness over other criteria. By conceptualizing a grid, the discretization of the Euclidean space, as a graph, we can encode the positions and states of all grid points and the coefficients of $f$ as the input features of the GNN. The concept of a grid as a graph allows us to extend the spatial domain of interest to the graph and apply the GNRK method to the coupled ODE system. Since the GNN utilizes shared parameters for both nodes and edges, the GNRK can efficiently train the state transitions from time $t$ to $t + \Delta t$ for all nodes, promoting data-efficient learning.

The GNRK represents a novel category of neural network-based PDE solvers, hitherto unexplored in previous literature. Its versatility extends beyond the examples showcased in this study, allowing it to be applied to a wide array of differential equations. Numerous avenues for further enhancement exist, such as implementing adaptive spatiotemporal discretization using intermediate results and addressing the challenge of solving equations based on noisy partial observations.

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

## A    DERIVATIVES IN NONUNIFORM SQUARE GRID

Since we have discretized Euclidean space into a nonuniform square grid, the derivatives of the governing equation $f$ need to be computed with care. We used the numerical differentiation method proposed in Sundqvist & Veronis (1970).

Consider a function $f$ with a 1D Euclidean spatial domain discretized by a nonuniform grid. We first compute the following half-derivatives at each grid point:

$$f'_{i-1/2} = \frac{f_i - f_{i-1}}{dx_{i-1}}, \quad f'_{i+1/2} = \frac{f_{i+1} - f_i}{dx_i}. \tag{9}$$

Then, we use them to compute the first and second derivatives of the $i$-th node as follows:

$$f'_i = \frac{dx_i}{dx_{i-1} + dx_i} f'_{i-1/2} + \frac{dx_{i-1}}{dx_{i-1} + dx_i} f'_{i+1/2}, \quad f''_i = \frac{f'_{i+1/2} - f'_{i-1/2}}{(dx_{i-1} + dx_i)/2}. \tag{10}$$

For a function $f$ defined in two or higher dimensions, the same calculation is performed independently for each axis.

## B    EXPERIMENT WITH THE 2D BURGERS' EQUATION

$$\frac{\partial u}{\partial t} = -u\frac{\partial u}{\partial x} - v\frac{\partial u}{\partial y} + \nu\left(\frac{\partial^2 u}{\partial x^2} + \frac{\partial^2 u}{\partial y^2}\right), \quad \frac{\partial v}{\partial t} = -u\frac{\partial v}{\partial x} - v\frac{\partial v}{\partial y} + \nu\left(\frac{\partial^2 v}{\partial x^2} + \frac{\partial^2 v}{\partial y^2}\right). \tag{11}$$

We consider the 2D Burgers' equation in spatial domin $\Omega \equiv [0,1]^2$ with periodic BC, and the time domain $\mathcal{T} \equiv [0,1]$. Initial conditions of both $u$ and $v$ are given as a 2D asymmetric sine function, $A\sin(2\pi x - \phi_x)\sin(2\pi y - \phi_y)\exp(-(x - x_0)^2 - (y - y_0)^2)$. In this context, $A$ serves as the normalization constant, ensuring a peak value of 1. Additionally, $\phi_x$ and $\phi_y$ represent the initial phases along each axis, and $x_0$ and $y_0$ denote the offsets accounting for any asymmetry. The constant coefficient $C$ corresponds to the viscosity parameter $\nu$, which is regarded as a global feature as it uniformly impacts the state of all nodes.

**Dataset.** All datasets adhere to the default initial condition settings, which entail $\phi_x = \phi_y = 0$ and $x_0 = y_0 = 0.5$. The coefficient $\nu$ remains fixed at 0.01, the grid size is uniformly set to $N_x = N_y = 100$, a uniform time step of $M = 1000$ is employed, and the simualation relies on a 4th-order Runge-Kutta method. Variability within each dataset is introduced through randomization of the values associated with the target regime, as described below.

- *Dataset I* evaluates the generalization performance with randomly selected initial conditions taken from $\phi_x, \phi_y \in (-\pi, \pi]$ and $x_0, y_0 \in (0, 1)$.

- *Dataset II* explores a range of values for $\nu$, spanning from 0.005 to 0.02, to assess the model's resilience to variations in the PDE coefficient $C$. When $\nu$ exceeds this specified range, the system experiences such significant dissipation that it rapidly reaches equilibrium within a fraction of a second. Conversely, if $\nu$ falls below this range, it leads to the generation of substantial shockwaves, which in turn can pose numerical stability challenges.

- *Dataset III* is constructed to validate the model's invariance to spatial discretization. It employs nonuniform square grids with dimensions $N_x$ and $N_y$ ranging from 50 to 150, resulting in varying total grid point counts within the range of 2,500 to 22,500. The spacing between these grid points deviates by approximately $\pm 10\%$ from that of the corresponding uniform grids of the same size.

- *Dataset IV* comprises samples generated using nonuniform temporal discretization denoted as $\{\Delta t_k\}_{k=1}^M$ from $k = 1$ to $M$, with a total of $M = 1000$ time steps. The difference in $\Delta t$, compared to the uniform case, fluctuates by approximately $\pm 10\%$. Furthermore, there is a distinction in trajectory quality between the training and testing sets, with $m = 1$ employed for training and $m = 4$ for testing, respectively.

**Grid-to-Graph transformation.** Since the governing equation 11 does not directly depend on the coordinate of each node, the position of each node is encoded as edge features representing its relative distance and direction to neighbors. Specifically, each edge is assigned a two-dimensional feature with the following rules. An edge of length $dx$ parallel to the $x$-axis has feature $[dx, 0]$, while an edge of length $dy$ parallel to the $y$-axis has feature $[0, dy]$.

**Implementation details for GNRK.** The embedding dimension of the input node, edge, and global features of $f_\theta$ are all 8. These encoded features are processed through the concatenated operations of the two GN modules shown in Fig. 2(b). In other words, equation 4 is repeated twice, which allows $f_\theta$ to compute the derivatives and Laplacians of equation 11. The four update functions are all 2-layer MLP with hidden layer size of 32 using GeLU activations and sum aggregation is used in both modules. This configuration results in the GNRK model having a total of 10,882 trainable parameters. Training the GNRK model on all datasets is accomplished using the AdamW optimizer. The learning rate is initially set at 0.004 for the first 20 epochs, reduced to 0.002 for the subsequent 30 epochs, and further decreased to 0.001 until reaching epoch 500. In each dataset, 20 random trajectories are provided for training the GNRK model, and the results presented in the main text are based on 50 different samples. It is important to note that GNRK is set to $m = 1$ only when training on *Dataset IV*; for all other cases, $m = 4$ is used.

**Implementation details for PINN.** We train the PINN, which is comprised of 6 hidden layers incorporating GeLU activations and batch normalization. The optimization objective is to minimize the total loss, which encompasses the PDE, initial, and boundary losses. We employ the AdamW optimizer with a learning rate of 0.0001 and utilize a cosine annealing scheduler as described in equation 12 to adjust the learning rate during training.

**Implementation details for FNO.** Implementation of FNO was performed with reference to (https://github.com/neuraloperator/neuraloperator). We train a 4-layer FNO with GeLU activation, 12 Fourier modes, and a width of 32 units. For optimization, we employ the AdamW optimizer with a learning rate set to 0.001, and we incorporate a cosine annealing scheduler, as detailed in equation 12. Our training approach for FNO is heuristic, where we extend the input channel by including coupled fields and PDE coefficients.

**Implementation details for GNO.** Implementation of GNO was carried out with reference to (https://github.com/neuraloperator/graph-pde). We train a 6-layer GNO with ReLU activation, with 64 width and 128 kernels. The AdamW optimizer with a learning rate 0.001 and a cosine annealing

scheduler in equation 12 are used. Edge attribution consists of the position values and two coupled field values in the 2D Burgers' equation at each node.

**Implementation details for GraphPDE.** Both the message network and aggregation network in the MPNN employed by GraphPDE consist of MLPs with a depth of 4, a width of 64, and utilize tanh activation functions. Like FNO, the GraphPDE is slightly modified from the original implementation to accomodate PDE coefficients as inputs. For the NeuralODE, which trains the MPNN by adjoint method, we utilize an adaptive Heun solver with an absolute tolerance set to $10^{-5}$. The GraphPDE model is trained using the AdamW optimizer with a learning rate of 0.0002 and a weight decay coefficient of 0.01.

## C  EXPERIMENT WITH COUPLED ODES

**Dataset.** For each of the three coupled ODE systems, we adopt a single dataset that encompasses four distinct criteria. Since each system exhibits unique demands in terms of initial conditions and coefficients, the specific requirements will be elaborated upon separately. In the spatial domain $\mathcal{G}$, the number of nodes and mean degrees are randomly selected from the intervals $[50, 150]$ and $[2, 6]$, respectively. As discussed in the main text, the topology of these domains is randomly chosen from among RR, ER, and BA network types. The simulation time varies for each system; nevertheless, for all scenarios, we employ nonuniform discretization with a time interval differing by approximately $\pm 10\%$ from the uniform case. Concerning the choice of the order $m$ for the Runge-Kutta method, similar to *Dataset IV* in Euclidean space, we employ low-quality $m = 1$ data for training and high-quality $m = 4$ data for evaluation.

**Implementation details for GNRK.** In all three systems, $f_\theta$ has the same structure, as shown in Fig. 2(b). Specifically, the GNRK uses a single GN module with 2-layer MLP update functions with GeLU activations and sum aggregation. As in the case of *Dataset IV* of the 2D Burgers' equation, we use a GNRK with $m = 1$ for training and modify it to $m = 4$ for testing. The model is trained using the AdamW optimizer, but the learning rates are different for each system as described separately. The learning rate $\eta$ at epoch $t$ follows a cosine annealing schedule as outlined below:

$$\eta(t) = \eta_{\min} + \frac{1}{2}(\eta_{\max} - \eta_{\min})\left(1 + \cos\frac{t}{T}\pi\right). \tag{12}$$

Whenever $t$ reaches the period $T$, both the maximum learning rate $\eta_{\max}$ and $T$ are adjusted by a predetermined factor.

### C.1  HEAT EQUATION

$$\frac{dT_i}{dt} = \sum_{j \in \mathcal{N}(i)} D_{ij}(T_j - T_i). \tag{13}$$

The initial conditions involve random assignment of each node to either a hot state ($T_i = 1$) or a cold state ($T_i = 0$), with the ratio of these two states also being randomly determined. The coefficients in this equation are represented by the edge feature $D_{ij} \in [0.1, 1.0]$. The heat equation is simulated for 2 seconds using $M = 100$ nonuniform time steps.

The node and edge embedding dimension is 16, and the hidden layer sizes at both per-edge and per-node update functions are 64. This results in a total of 7,201 trainable parameters. Regarding the learning rate, we use $\eta_{\min} = 0.0001, \eta_{\max} = 0.01$, and $T = 10$. At each occurrence of $t$ reaching the period $T$, the period $T$ doubles, and $\eta_{\max}$ decreases by a factor of 0.7, continuing until reaching epoch 630.

### C.2  KURAMOTO EQUATION

$$\frac{d\theta_i}{dt} = \omega_i + \sum_{j \in \mathcal{N}(i)} K_{ij}\sin(\theta_j - \theta_i). \tag{14}$$

Each node initializes the system with a random phase within the range of $(-\pi, \pi]$. One of the coefficients, denoted as $C$, represents the node feature $\omega_i$, which is randomly sampled from a normal

distribution with an average of 0 and a standard deviation of 1. The other $C$ corresponds to the edge feature $K_{ij}$. It is well known that large values of $K_{ij}$ can lead to a synchronized state in which all the oscillators have the same phase. Therefore, we limit the range of $K_{ij}$ to a relatively small interval of $[0.1, 0.5]$ to ensure that the system exhibits diverse trajectories without synchronization. We conduct a simulation lasting 10 seconds, employing $M = 500$ nonuniform time steps.

Since the state $\theta_i$ is periodic, we preprocess it by computing both the cosine and sine values. The pre-processed $\theta_i$ and node feature $\omega_i$ are embedded into 16 dimensions using their respective encoders, and $K_{ij}$ is also represented in 16 dimensions with the edge encoder. Both update functions $\phi^e$ and $\phi^v$ have hidden dimensions with a size of 64, resulting in a total of 16,097 trainable parameters. The learning rate varies according to equation 12, where $\eta_{\min} = 0.00001, \eta_{\max} = 0.01$, and $T = 10$ are initially used. For every cycle of the cosine schedule, $T$ is doubled and $\eta_{\max}$ is halved, for a total of 310 epochs.

### C.3 COUPLED RÖSSLER EQUATION

$$\frac{dx_i}{dt} = -y_i - z_i, \quad \frac{dy_i}{dt} = x_i + ay_i + \sum_{j \in \mathcal{N}(i)} K_{ij}(y_j - y_i), \quad \frac{dz_i}{dt} = b + z_i(x_i - c). \quad (15)$$

The initial condition is established by randomly selecting $x_i, y_i \in [-4, 4]$ and $z_i \in [0, 6]$ for each node, although it is worth noting that the trajectory can travel outside this range. The coefficients governing the equations include the global features $a, b, c$ and the edge features $K_{ij}$. It is widely known that the system can exhibit chaos depending on the values of $a, b$, and $c$. To capture chaotic trajectories, we restrict $a, b \in [0.1, 0.3]$ and $c \in [5, 7]$. Similar to the Kuramoto system, we choose $K_{ij} \in [0.02, 0.04]$ to prevent synchronization, ensuring that the positions of all nodes evolve indi-vidually. We track the trajectory for 40 seconds with $M = 2000$ nonuniform time steps.

Three separate encoders are used for each set of coordinates $(x, y, z)$ of the nodes, with each encoder producing 32-dimensional node features. Similarly, each of the coefficients $a, b, c$ and $K_{ij}$ is associated with a dedicated encoder generating 32-dimensional embeddings. Leaveraging these embedded features, two update functions $\phi^e$ and $\phi^v$ with 128 hidden units complete the GN module. The total number of trainable parameters for the GNRK model amounts to 79,267. The learning rate follows equation 12 as before, initially set to $\eta_{\min} = 0.00001, \eta_{\max} = 0.002$, and $T = 20$. In each cycle, up to a total of 2,000 epochs, $T$ increases by a factor of 1.4, and $\eta_{\max}$ decreases by a factor of 0.6.

