# OpenReview forum: "GNRK: Graph Neural Runge-Kutta method for solving partial differential equations"
_ICLR.cc/2024/Conference — Submitted to ICLR 2024_

### Official Review · Reviewer_pwWA · 2023-10-30

**Soundness:** 2 fair
**Presentation:** 3 good
**Contribution:** 2 fair
**Rating:** 5
**Confidence:** 3

**Summary:**

The paper presents a neural network model combined with the explicit Runge-Kutta time integration scheme to solve some 1D/2D ODE and PDE problems. For PDE problems, the network model essentially fits the (discretized) spatial gradient operators in the PDE.

**Strengths:**

At a high level, I appreciate the combination of classical numerical methods with neural network models. I think this is a right move that deserves more attention in the community.

I also like that the paper puts thoughts on network generalization and explicitly discusses multiple scenarios: different initial conditions, spatial discretizations, PDE coefficients, etc.

**Weaknesses:**

I struggle to see a motivation for using a neural network model to replace the evaluation of f in Eqn. (1) in the RK framework. The RK scheme described by Eqn. (3) is a standard explicit time integration scheme, and a numerical method (finite difference, finite elements, etc.) for evaluating f in Eqn. (1) can be highly efficient using trivial parallelization on GPUs. Numerical methods also have great generalizability as they can easily handle different initial conditions, physical parameters, spatial discretization, etc. in a principled way. After reading the paper, it remains unclear to me what the advantages of the proposed method over numerical simulation are.

I also have a number of concerns regarding the experiments:
- Following my comment above, I feel the method should be compared with a high-quality numerical simulator (finite difference/element/volume, etc.) and report their runtime difference. I skimmed over the Burges equation part in the code, and it looks like the training data were generated using finite differences implemented in numpy. Was it faster or slower than the network inference time in the proposed method?
- I feel the comparison with other neural network baselines is not well-balanced in that it exaggerates their weaknesses but ignores their strengths in other aspects. Taking PINN as an example, PINN has been tested against much more complicated PDEs in fluid and solid mechanics (e.g., Navier-Stokes) which explicit time integrations like RK4 struggle to solve, and I don’t see an easy way for extending the proposed method to these PDEs given that it is coupled with RK time integration. Another thing is that PINN can be trained with partial observations of the spatiotemporal states sampled irregularly in time and space, so it seems to impose fewer requirements on the training data than the proposed method. Of course, it is unnecessary to beat all baselines in all aspects before publishing a research paper, but I think a more balanced view and experiment design would make it more convincing.

**Questions:**

I actually don’t have any technical questions to ask for now. The paper is well-written, and the main method is quite straightforward to follow. Maybe I will ask some technical questions after reading the rebuttal.

---

> ### Author Response · Authors · 2023-11-17
> **Response to reviewer pwWA**
>
> Here, we will organize and respond to individual comments that were not addressed in general response.
>
> 1. I struggle to see a motivation for using a neural network model to replace the evaluation of $f$ in Eqn. (1) in the RK framework.
>
> $\Rightarrow$ The advantage of replacing $f$ with a neural network in the GNRK lies in scenarios where the exact form of the governing equation is unknown.
> In cases where the equation $f$ is not precisely defined, numerical methods may face challenges, while the GNRK can be trained using data alone, without prior knowledge of the governing equation. This flexibility allows the proposed method to adapt to situations where traditional numerical simulations struggle due to the lack of a well-defined analytical form for the function $f$.
>
> 2. I feel the method should be compared with a high-quality numerical simulator (finite difference/element/volume, etc.) and report their runtime difference.
>
> $\Rightarrow$ In response, we have included the wall clock and GPU memory usage for the GNRK and other baseline models in Table. 1 for a more comprehensive comparison.
> Regarding the implementation of numerical simulations, we acknowledge the use of a naive finite difference method with numpy, without GPU acceleration or mroe optimized methods due to the time constraints.
> While the GNRK may exhibit slower wall clock times compared to typical numerical solvers, it offers a unique advantage in situations where little information about the governing equation is available, as highlighted in the previous response.
>
> 3. I feel the comparison with other neural network baselines is not well-balanced in that it exaggerates their weaknesses but ignores their strengths in other aspects. Taking PINN as an example, PINN has been tested against much more complicated PDEs in fluid and solid mechanics (e.g., Navier-Stokes) which explicit time integrations like RK4 struggle to solve, and I don’t see an easy way for extending the proposed method to these PDEs given that it is coupled with RK time integration.
>
> $\Rightarrow$ As you pointed out, for chaotic systems such as Navier-Stokes, error can accumulate when using recursive time integration.
> In the case of the PINN, where no time integration is involved, it can predict arbitrary time without accumulating error.
> However, applying the PINN to such complex PDEs, especially those with coupled dynamics like 2D Burgers' equation, presents challenges.
> Training a neural network to predict the solution directly, as PINN does, becomes intricate, especially as the spatial domain dimension increases.
> The exponential increment in required training points for PINN with higher dimensions, along with challenges in balancing data points at boundaries and non-boundary regions, makes it less straightforward.
> Moreover, it is not applicable to Dataset I (different initial conditions), and Dataset II (different PDE coefficients)
>
> 4. Another thing is that PINN can be trained with partial observations of the spatiotemporal states sampled irregularly in time and space, so it seems to impose fewer requirements on the training data than the proposed method.
>
> $\Rightarrow$ Indeed, a significant advantage of PINN lies in its ability to handle arbitrary points in both spatial and temporal domains.
> It is demonstrated in our experiments using Dataset III (nonuniform spatial domain) and IV (nonuniform temporal discretization).
> However, it's essential to acknowledge that optimizing PINN for challenging problems like 2D Burgers' equation can be intricate, potentially leading to less accurate predictions.

---

> > ### Comment · Reviewer_pwWA · 2023-11-22
> > **Thank you for the reply**
> >
> > Thank you for your clarification. While I remain concerned with the submission, I buy the claim about the usage of network methods and numerical simulations when the governing equation is unknown, so I will raise my score to 5.

---

### Official Review · Reviewer_r1o4 · 2023-11-01

**Soundness:** 3 good
**Presentation:** 2 fair
**Contribution:** 2 fair
**Rating:** 3
**Confidence:** 4

**Summary:**

This paper introduces a novel model, Graph Neural Runge-Kutta (GNRK), which combines classical numerical solvers with graph neural networks. GNRK leverages Runge-Kutta to address time-dependent partial differential equations (PDEs) and is designed to be generally applicable under varying conditions, including changes in initial conditions, PDE coefficients, and modifications in time and space discretization.

# After rebuttal!
Thank you for addressing my inquiries about the paper. I appreciate your thoughtful responses, and I hope that my questions and suggestions have been beneficial for the paper. Additionally, there are aspects I'd like to see included in the future, such as experiments related to question 8, specifically showcasing experiments involving nu. Furthermore, it would be valuable to have comparisons with other baselines that you mentioned haven't been conducted yet due to time constraints. I believe that addressing these aspects would significantly enhance the paper. Currently, it feels a bit incomplete without these additions, and I hope to maintain the score I have given, expecting improvements in the future. Thank you.

**Strengths:**

Research that combines deep learning techniques like graph neural networks and message passing with well-established numerical analysis methods is of significant importance. This integration of mathematical principles and deep learning technologies has the potential to create more robust and effective models.

**Weaknesses:**

Firstly, the title of the paper includes "PDE," but the experimental results mainly focus on 2D Burgers' equations as PDE, while all other experiments seem to center on coupled ODE systems. Also, the phrase in the abstract, "often confined to specific PDEs under certain constraints," seems somewhat unclear in its meaning. There are different numerical schemes (e.g., FDM, FEM, or more specialized variations) for applying numerical analysis methods to different types of equations. It appears that neural network models have various methods for different types of PDEs. More detailed explanations in the paper title and abstract might be needed to better represent this research.

Secondly, the paper lacks a clear explanation of how the novelty of the GNRK model differs from existing models. It is essential to clarify where GNRK offers advantages over previous models concerning the combination of numerical analysis and graph neural networks. The novelty mentioned on page 2 and the four categories discussed in Section 3 require a precise comparison with prior research. For instance, the advantage related to the initial condition might not be unique to GNRK but rather a common benefit shared with all models utilizing graph neural networks. Additionally, the statement that learning with changing PDE coefficients is not efficient, as mentioned in Section 3, should be supported by evidence. In light of this, focusing on the advantages of GNRK over other neural network models, considering multiple aspects of operator learning models and models that learn solutions varying over time, and creating a table that clearly illustrates what each model can and cannot do would be beneficial. This table could emphasize where GNRK excels in comparison.

Furthermore, there appears to be a lack of mention and references to several recent papers that use graph neural networks for simulating physical phenomena. In fact, papers [1], [2], [3], [4], for instance, utilize GNNs to predict changes between the previous and subsequent time steps and consider time-dependent predictions. It would be helpful to discuss the commonalities and differences between these papers and GNRK, particularly [1] and [2], which deal with time-dependent PDEs, and explain how they relate to the work in this paper. Additionally, considering additional types of data in this paper could further highlight the advantages of GNRK.

*[1] Brandstetter, J., Worrall, D., & Welling, M. (2022). Message passing neural PDE solvers. arXiv preprint arXiv:2202.03376.*

*[2] Boussif, O., Bengio, Y., Benabbou, L., & Assouline, D. (2022). MAgnet: Mesh agnostic neural PDE solver. Advances in Neural Information Processing Systems, 35, 31972-31985.*

*[3] Pfaff, T., Fortunato, M., Sanchez-Gonzalez, A., & Battaglia, P. W. (2020). Learning mesh-based simulation with graph networks. arXiv preprint arXiv:2010.03409.*

*[4] Lienen, M., & Günnemann, S. (2022). Learning the dynamics of physical systems from sparse observations with finite element networks. arXiv preprint arXiv:2203.08852.*

**Questions:**

* More detailed explanation is needed for the phrase 'less dependent on the form of PDEs or conditions.' at page1. What does it mean?

* In Section 2.1, it would be helpful to clarify how GNRK can handle non-uniform discretization. For example, if there are data points at 1 second, 2 seconds, and 5 seconds, how can GNRK capture the information that the intervals between 1 and 2 seconds and between 2 and 5 seconds are different?

* In the middle of page 3, there is an assumption about "same degree." Is this assumption absolutely necessary, or can GNRK be applied without it? You've clarified that GNRK is applicable in cases like the left graph in Figure 1, but what about scenarios like the right graph in Figure 1? Can GNRK be used there as well?

* Regarding Figure 2, when determining the order of Runge-Kutta and constructing the model, can GNRK be applied to predict solutions only at that order?

* In the Burgers experiments in Section 4.1, it would be interesting to know how the model in Figure 2 adapts and learns GNRK as nu changes. This is one of the novelties mentioned in Section 3 but is not detailed in the main text.

* Section 4.2 discusses experiments related to coupled ODEs without comparative benchmarks. Could you explain why there are no benchmarks for coupled ODEs? Is it possible to apply the benchmarks in Table 1 to coupled ODEs, or are there other neural network models suitable for coupled ODEs that are worth mentioning?

---

> ### Author Response · Authors · 2023-11-17
> **Response to reviewer r1o4**
>
> Here, we will organize and respond to individual comments that were not addressed in general response.
>
> 1. More detailed explanations in the paper title and abstract might be needed to better represent this research.
>
> $\Rightarrow$ In response to your feedback, we have refined the the second sentence of abstract for better clarification.
> > However, they may perform poorly or even be inapplicable when deviating from the conditions for which the models are trained, in contrast to classical PDE solvers that rely on numerical differentiation.
>
> We hope this modification enhances the clarity and better represents the research focus.
>
> 2. The paper lacks a clear explanation of how the novelty of the GNRK model differs from existing models.
>
> $\Rightarrow$ We appreciate your attention to the novelty of the GNRK and the need for a clearer explanation of its advantages over existing models.
> However, there seems to be a slight misunderstanding regarding the Table. 1, which outlines the applicability and performance of various approaches.
> We would like to clarify that the '-' in the table indicates instances where a model cannot be applied to a particular dataset.
>
> 3. Lack of mention and references to several recent papers that use graph neural networks for simulating physical phenomena.
>
> $\Rightarrow$ Thank you for highlighting the important landmark papers relevant to current works.
> While we acknowledge the relevance of the papers, we were constrained by time and could not conduct experiments to include them in our discussion.
>
> 4. More detailed explanation is needed for the phrase 'less dependent on the form of PDEs or conditions.' at page1. What does it mean?
>
> $\Rightarrow$ We have modified the sentence to explicitly state that classical solvers like FEM, FDM are generic, which can find the solution of a PDE regardless of its form, initial and boundary conditions.
> > Unlike NN solvers, classical solvers such as finite-element method (FEM), finite-difference method (FDM), and finite-volume method (FVM) can find the solution of the PDE regardless of its form, initial and boundary conditions, given an exact PDE.
>
> 5. It would be helpful to clarify how GNRK can handle non-uniform discretization
>
> $\Rightarrow$ To clarify, the simulation starts with a pre-defined randomized $\{ \Delta t \}$.
> The detailed explanation of how GNRK captures nonuniform time intervals is provided in Section 3.
> We highlight that the GNRK captures time information using $\Delta t$.
> Specifically, $\Delta t$ affects the Runge-Kutta scheme, as illustrated in Fig. 2(a), while the trainable neural network $f_\theta$ remains independent of $\Delta t$.
>
> 6. In the middle of page 3, there is an assumption about "same degree." Is this assumption absolutely necessary, or can GNRK be applied without it?
>
> $\Rightarrow$ In short, the assumption of 'same degree' is not necessary for the GNRK.
> GNRK can be applied to scenarios where the degrees of nodes are different, as illustrated in the right plot of Fig. 1.
> The term `grid' refers to a special case where all degrees are equal, as shown in left plot of Fig. 1.
> We have presented experimental results on general graphs, which has varying degrees, in Section 4.2.
>
> 7.  Regarding Figure 2, when determining the order of Runge-Kutta and constructing the model, can GNRK be applied to predict solutions only at that order?
>
> $\Rightarrow$ GNRK can be applied to different orders without retraining.
> The trainable part in the GNRK is represented by $f_\theta$ in Fig. 2(b), and it is recurrently reused as in Fig. 2(a).
> For instance, the GNRK with an order of 1 can be used for training, and the trained $f_\theta$ can be employed for prediction by modifying the structure of the GNRK to have an order of 4.
> This approach is utilized for Dataset IV in Section 4.1 and Section 4.2.
>
> 8. In the Burgers experiments in Section 4.1, it would be interesting to know how the model in Figure 2 adapts and learns GNRK as nu changes. This is one of the novelties mentioned in Section 3 but is not detailed in the main text.
>
> $\Rightarrow$ In the GNRK model, $\nu$ is included as one of the input features.
> Therefore, as $\nu$ changes, the structure of the GNRK remains constant.
> The model adapts to variations in $\nu$ by adjusting the purple $\mathbf{g}$ component in Fig 2(b).

---

### Official Review · Reviewer_yG9N · 2023-11-06

**Soundness:** 3 good
**Presentation:** 3 good
**Contribution:** 3 good
**Rating:** 3
**Confidence:** 4

**Summary:**

# Initial review summary
In this paper, the authors propose Graph Neural Runge-Kutta method, which utilizes graph networks to capture the spatial structure, and the Runge-Kutta method for the temporal integration. It is claimed that the proposed model can better handle general PDE problems. Experiments in Burgers' equation have been performed, in comparison with baseline methods such as FNO, GNO, GraphPDE, etc.

# Update after rebuttal

Thanks very much for your detailed response. Some of my questions are definitely clarified. It is also very inspiring to go through other reviewers' questions and replies.

- I am still concerned if it is necessary or beneficial to choose $a$ and $b$ strictly from the Butcher's table, while $f$ itself has lost its rigorousness.
- Without the capability of remeshing, I think there will be a gap between the proposed method and practical usage, since it will face time-dependent PDE problems where fixed mesh has limits. Also, I don't think it is trivial to integrate remeshing into the proposed method.
- Other reviewers' concerns about the experimental settings, evaluation metrics, comparison baselines, etc., are also very solid questions.

With all the factors considered, I tend to remain my original score for now.

**Strengths:**

- Finding the sweet point to combine data-driven models and numerical methods is an important and interesting topic in AI4PDE.
- Benefit from the design, the proposed method can estimate the solution at any position or time.

**Weaknesses:**

- There are a few popular works solving spatial-temporal PDEs with graph neural networks that are not mentioned or compared in the paper, such as meshgraphnet, MP-PDE, etc. There are also a few works in Neural ODEs that are relevant, such as hypersolver, deep Euler method, etc.
- Traditionally, the Runge-Kutta method comes with strict guarantees in accuracy, convergence, stability, etc. But with a GNN as the estimator in a timeslot, such theoretical properties are lost. So I wonder if it is still necessary or beneficial to utilize the Runge-Kutta scheme with GNN when handling a practical PDE problem.
- In the conclusion section, it is claimed that "it is invariant to spatial and temporal discretization". But I don't see how the proposed method can guarantee such invariance capability.

**Questions:**

- I am wondering which numerical method suits the situation better, the Runge-Kutta method or the multi-step method. And I would like to know if the authors have some comments on this.
- I don't see how remeshing is handled in the proposed method? Remeshing is a vital part of time-dependent PDEs, especially for fluid problems, hence I think remeshing should be a necessary part in the solution.

---

> ### Author Response · Authors · 2023-11-17
> **Response to reviewer yG9N**
>
> Here, we will organize and respond to individual comments that were not addressed in general response.
>
> 1. Traditionally, the Runge-Kutta method comes with strict guarantees in accuracy, convergence, stability, etc. But with a GNN as the estimator in a timeslot, such theoretical properties are lost. So I wonder if it is still necessary or beneficial to utilize the Runge-Kutta scheme with GNN when handling a practical PDE problem.
>
> $\Rightarrow$ Classical numerical solvers, including the Runge-Kutta method, provide rigorous guarantees when the governing equation is precisely known.
> The GNRK, on the other hand, offers a unique advantage by allowing training and prediction without prior knowledge of the PDE.
> This is particularly beneficial in scenarios where the exact governing equation is unkown, but relevant observational data and parameters are available.
> While the GNRK may not inherit the theoretical guarantees of classical solvers, its flexibility in handling unknown PDEs makes it a valuable tool in situations where classical solvers might face limitations.
>
> 2. In the conclusion section, it is claimed that "it is invariant to spatial and temporal discretization". But I don't see how the proposed method can guarantee such invariance capability.
>
> $\Rightarrow$ In respose, we modified the phrase to make it clearer as follows:
> `it can be applied to any arbitrary spatial and temporal discretization, allowing ...'
> The claim of invariance to spatial and temporal discretization is based on the inherent properties of the GNRK outlined in Section 3.
> - Spatial discretization: The GNRK handles spatial discretization by considering the coordinates of each grid point as input features for nodes or edges.
> The parameter sharing mechanism of GNN allows GNRK to effectively capture variations in these features, making it adaptable to different spatial discretization schemes.
> - Temporal discretization: The GNRK views time as the time interval $\Delta t$.
> Importantly, this temporal information only affects the Runge-Kutta scheme (Fig 2(a)), and not the trainable neural network $f_\theta$.
> Therefore, the GNRK remains accurate even when dealing with arbitrary $\Delta t$ in temporal discretization.
>
> 3. I am wondering which numerical method suits the situation better, the Runge-Kutta method or the multi-step method. And I would like to know if the authors have some comments on this.
>
> $\Rightarrow$ The choice between the Runge-Kutta method and multi-step method may depends on the specific characteristics of the problem being sovled.
> In the context of the paper, the Runge-Kutta method is used for a representative numerical solver.
> It is important to note that the GNRK approach is designed to be versatile, and there is potential for its application to other numerical solving methods.
> If you have a specific multi-step method in mind or if there are particular aspects you would like us to consider, please provide more details, and we'll be happy to discuss further.
>
> 4. I don't see how remeshing is handled in the proposed method?
>
> $\Rightarrow$ The current implementation of the proposed GNRK method does not explicitly incorporate time-dependent remeshing.
> The method operates on non-uniform grids that are initially chosen and remain fixed over time.
> Each prediction at a time step is made independently, allowing the GNRK to handle predictions on different grids.
> Therefore, the GNRK can potentially be combined with a remeshing technique and corresponding interpolation scheme.
> Integrating time-dependent adaptive remeshing is an interesting direction for future research, as it could enhance the accuracy and applicability of the GNRK, especially in scenarios with dynamic or evolving geometries.

---

### Official Review · Reviewer_Sxmd · 2023-11-09

**Soundness:** 4 excellent
**Presentation:** 4 excellent
**Contribution:** 3 good
**Rating:** 5
**Confidence:** 5

**Summary:**

A novel surrogate model based on graph neural networks (GNNs) to solve partial differential equations (PDEs) is introduced. The proposed model is endowed with a recurrent structure to mimic numeral solvers. It is achieved by integrating Runge-Kutta scheme. This model can handle different PDE conditions.

**Strengths:**

1/ Originality and significance:

While GNNs become the standard models to approximate the functional space of PDEs, integrating Runge-Kutta scheme to GNNs is a key contribution. As a consequence, GNN based on Runge-kutta scheme enables tackling PDE problems under different conditions and can enjoy good generalization capabilities where existing GNN based PDEs fail.

2/ Quality:

The proposed work is interesting for the community, it opens new perspective at the crossroad of numerical solvers and neural networks. The motivation to  generalize to different PDE conditions is well structured, the claims are well supported in the experiments.
The claims on the ability to cope with different initial conditions, PDE coefficients and spatio-temporal discretization are important for engineers in real-world applications

3/Clarity:

The paper is well structured and easy to follow.
Discussion of related works is sufficient.
Results are reproducible and the code is easy to follow.

**Weaknesses:**

One of the weaknesses of this work is in its experimental protocol. It has not been validates on large scale datasets and problems with challenging geometries and high reynolds(10^6).
The value of the paper will be important if it is validated on challenging problem, proving that the following claims still hold at large scale:
• Initial conditions
• PDE coefficients
• Spatial discretization
• Temporal discretization and the order of RK

**Questions:**

1/ It is mentioned in the paper "It can even generate solutions with higher precision than the training data through the RK order adjustment":
Can you explain more how it is possible ? is it just inference or fine-tuning at test time

2/ It would be very important to validate the proposed work on challenging datasets. For instance:
A- EAGLE: Large scale learning of turbulent fluid dynamics with mesh transformers.  About 1 million 2D meshed with 600 different scenes
B- AirfRANS: High Fidelity Computational Fluid Dynamics Dataset for Approximating Reynolds-Averaged Navier–Stokes Solutions. Complex geometries, high Reynold numbers and challenging to obtain accurate results at the boundary layer.
C- PDEBench: An Extensive Benchmark for Scientific Machine Learning. For instance considering the following datasets: 3D_cfd, 2D_cfd, darcy_flow.

I would be happy to increase my score if the suggested new experiments validate the claims of the paper.

**Details Of Ethics Concerns:**

No ethics concerns

---

> ### Author Response · Authors · 2023-11-17
> **Response to reviewer Sxmd**
>
> Here, we will organize and respond to individual comments that were not addressed in general response.
>
> 1. It has not been validates on large scale datasets and problems with challenging geometries and high reynolds($10^6$)
>
> $\Rightarrow$ Thank you for your observation regarding in the experiment protocol, particularly the absence of validation on large-scale datasets and challenges with complex geometries and high reynolds.
> We acknowledge this limitation.
> The focus of this study is to introduce a new approach to solving PDEs, and we chose relatively simple problems for the initial comparison.
> While we recognize the importance of validating on more complex and large-scale datasets, we hope this work lays the groundwork for future researches in that direction.
>
> 2. Can you explain more how 'generating higher precision than the training data' is possible?
>
> $\Rightarrow$ For clarification, we rephrased the sentence as follows:
> > It can even predict solutions with higher precision than the training data by modifying its' structure without retraining or fine-tuning.
>
> During the training phase, the GNRK is trained with a specific order, and the trained $f_\theta$ captures the underlying dynamics.
> However, at test time, the GNRK can adapt to different orders by changing the recurrent number in Fig 2(a), without requiring additional training.
> This flexibility allows us to generate solutions with higher precision than the training data.
> For instance, if the model is trained with an Runge-Kutta order of 1, it can be adjusted to order 4 at the test time, enabling more accurate predictions.
> The result of this order adjustment capability is shown in Dataset IV in Section 4.1 and Section 4.2.
>
> 3. It would be very important to validate the proposed work on challenging datasets.
>
> $\Rightarrow$ We appreciate your suggestions for challenging datasets, particularly in fluid dynamics research.
> As in general response, it's essential to note that the primary focus of this work is on introducing novel approach for AI PDE solvers rather than adapting existing approaches to specific PDEs.
> The experiment on 2D Burgers' equation is conducted on a relatively small scale.
> The training involves 20 trajectories generated from 10,000 2D grid points, and predictions were made for 50 different trajectories.
> While this may not cover the complexities of the large-scale datasets you suggested, it provides a foundation for future exploration and validation on more challenging problems.
> Although, your specific suggestions are valuable, and the consideration of such datasets could be a direction for further research to assess the scalability and robustness of the proposed GNRK approach.

---

> > ### Comment · Reviewer_Sxmd · 2023-11-30
> > **Review follow-up**
> >
> > I thank the authors for providing detailed explanations. However, l still think that it's important for the community to have some results on more challenging (and large scale) datasets. Thus, l keep my original score.

---

### Official Review · Reviewer_MKmy · 2023-11-10

**Soundness:** 1 poor
**Presentation:** 2 fair
**Contribution:** 2 fair
**Rating:** 3
**Confidence:** 4

**Summary:**

The paper proposes the GNRK as a PDE solver by using graph neural networks as the message passing mechanism, which simulates the multi-step Runge-Kutta method with a recurrent structure. The proposed method can handle nonuniform spatial grids and temporal intervals by learning graph representations from node and edge, and global information. GNRK can be trained with a small amount of data, and outperforms the classical neural operators and PINNs in 2D Burgers' Equation. Moreover, experiments in three datasets of the heat equation, Kuramoto equation, and coupled Rossler equation proved that the fourth-order solver model outperforms the first-order solver model, and the network can handle three types of graph topologies well.

**Strengths:**

- The design in GNRK is reasonable. Directly modeling observation time with Runge-Kutta method enables the model to deal with nonuniform observation time intervals. Incorporating graph neural network enables models to cope with irregular meshes.

- GNRK beats other baselines in the 2D Burgers' Equation dataset with less parameters.

**Weaknesses:**

1. There exists serious mismatches between the authors' claim and their experiments.

The authors claim that GNRK can be generalized to different initial conditions, PDE coefficients, spatial discretization and temporal discretizaition without retraining in Section 3. However, **no experiments are provided to prove the generalization without retraining**. They just change the Runge-Kutta order between train and test datasets in each experiment. For me, to demonstrate the model generality, they should train the model in Dataset I and test in Dataset 2.

2. About the experiment settings.

The experimental datasets are too simple to provide converncing results.

- In Appendix B, the initialized state of the 2D Burgers' equation is determined by a quite simple periodic function, which only changes the phase and the center point. Reference to FNO[3], the 2D Burgers' Equation dataset can be initialized using a normal distribution.
- Besides, for three datasets mentioned in section 4.2, they only experiment GNRK and do not provide any comparisons to other baselines. Thus, it is hard to judge the model performance in irregular meshes.

3. About the model efficiency.

Higher order of Runge-Kutta leads to higher computational cost, the comparisions between GNRK and other baselines in running time, GPU memory are expected.

4. About the baselines and ablations.

FNO, GNO, GraphPDE, PINN are both not the SOTA baselines. Recently, many advanced neural operators have been proposed, such as LSM [1] and geo-FNO [2]. They can also handle the irregular input meshes. How about directing comparing with them or replacing the GNN parts with them?


```
[1] Solving High-Dimensional PDEs with Latent Spectral Models, ICML 2023
[2] Fourier Neural Operator with Learned Deformations for PDEs on General Geometries, arXiv 2022
```

5. About the showcases.

The authors only provide the showcases for their own model. More showcase comparisons are expected to intuitively demonstrate the effectiveness of GNRK.

**Questions:**

The design of the network seems reasonable. But the experiments can not verify the performance of the proposed model. All the details can be found in the weakness section.

---

> ### Author Response · Authors · 2023-11-17
> **Response to reviewer Mkmy**
>
> Here, we will organize and respond to individual comments that were not addressed in general response.
>
> 1. There exists serious mismatches between the authors' claim and their experiments.
>
> $\Rightarrow$ As you highlighted, our assertion is that the GNRK can be generalized to diverse initial conditions (ICs), PDE coefficients, or other variables without retraining. Table 1 displays the test error for each dataset, where training was conducted under specific conditions. For example, Dataset I comprises trajectories originating from various ICs, aiming to evaluate the model's adaptability to ICs beyond those used in training. We intend to revise the main text to enhance clarity and prevent any potential misunderstandings arising from this presentation.
> The scenario you propose, where the GNRK is trained on Dataset I (different ICs, same coefficients) and subsequently tested on Dataset II (same ICs, different coefficients), poses a formidable challenge. This is because the model wasn't exposed to any information regarding the varying coefficients in Dataset II during its training phase.
>
> 2.  In Appendix B, the initialized state of the 2D Burgers' equation is determined by a quite simple periodic function
>
> $\Rightarrow$ The chosen initial condition is expressed as $A \sin(2\pi x - \phi_x) \sin(2\pi y - \phi_y) \exp(-(x-x_0)^2 - (y-y_0)^2)$.
> It's noteworthy that this formulation introduces changes not just in phase (i.e., the peak locations of the $sin$ function) but also includes asymmetry in the curvature of each peak.
> Furthermore, it exhibits steeper slopes along the $x$ and $y$ axes compared to a normal distribution.
> We believe these characteristics add complexity to solving the 2D Burgers' equation.
>
> 3. Higher order of Runge-Kutta leads to higher computational cost, the comparisions between GNRK and other baselines in running time, GPU memory are expected.
>
> $\Rightarrow$ We acknowledge that a higher order of Runge-Kutta could potentially result in increased computational cost.
> To address this concern, we have included additional information in Table. 1, introducing two new rows specifically for the wall clock time and GPU memory usage required for predicting a single trajectory.

---

> > ### Comment · Reviewer_MKmy · 2023-11-22
> > **Thanks for your response**
> >
> > Thanks for the author's response and effort in providing the efficiency comparison. I still have concerns with the usage of "generalization" and insufficient baselines. Thus, I would like to keep my original score.

---

### Author Response · Authors · 2023-11-17
**General response to reviewers**

Dear reviewers: Thank you for your valuable comments. We have addressed common questions in our response. Your insights are crucial for improving the manuscript. We are open to more feedback and discussion.

1. Lack of recent, SOTA benchmark models for 2D Burgers' equation

$\Rightarrow$  Our main objective is to showcase the promise of merging a versatile numerical method with expressive neural networks. Conducting practical comparisons with current state-of-the-art benchmark models presents an intriguing area for future research that we plan to explore.

2. 2D Burgers' equation is too simple task to showcase the capabilities of a PDE solver

$\Rightarrow$ We would like to highlight the challenge of solving the 2D Burgers' equation, which entails two interconnected fields, $u$ and $v$, in contrast to the more commonly utilized 1D Burgers' equation. Given its relative scarcity in literature, we believe that exploring this application can significantly contribute to the field of PDE solvers.

3.  No benchmarks for graph spatial domain

$\Rightarrow$ Handling coupling ODE systems in the graph spatial domain is to demonstrate the extensibility of the GNRK.
There are currently few models applicable to this less-explored system, but the papers that introduce the models have not specifically addressed these systems.

---

### Comment · Area_Chair_fpZM · 2023-11-23
**From AC at the end of rebuttal: Reviewer response required**

Dear Reviewers,

Thanks for your time and commitment to the ICLR 2024 review process.

As we approach the conclusion of the author-reviewer discussion period (Wednesday, Nov 22nd, AOE), I kindly urge those who haven't engaged with the authors' dedicated rebuttal to please take a moment to review their response and share your feedback, regardless of whether it alters your opinion of the paper.

Your feedback is essential to a thorough assessment of the submission.

Best regards,

AC

---

### Meta-Review · Area_Chair_fpZM · 2023-12-10

**Metareview:**

This paper solves partial differential equations with a Graph Neural Runge-Kutta (GNRK) method that integrates GNN modules with a recurrent structure inspired by the classical solvers. The claimed contributions are generalization to spatial and temporal discretizations, initial conditions or PDE coefficients. While the targeted properties are very desirable, they are not well approached by this study. After the rebuttal, all five reviewers are in unanimous consistency of rejection, grounding on the concerns of overclaiming (all cases shall be tested as claimed) and insufficient evaluation (more realistic settings, more competitive baselines and more difficult datasets are needed).

**Justification For Why Not Higher Score:**

The paper has significant flaws as a scientific study, in the forms of overclaiming and weak empirical study.

**Justification For Why Not Lower Score:**

N/A

---

### Decision · Program_Chairs · 2024-01-16

Reject